# OXSR1 inhibits inflammasome activation by limiting potassium efflux during mycobacterial infection

Elinor Hortle[1,2,3], Vi LT Tran[1], Kathryn Wright[1], Angela RM Fontaine[4], Natalia Pinello[5,6], Matthew B O'Rourke[3], Justin J-L Wong[5,6], Philip M Hansbro[3], Warwick J Britton[1,7], Stefan H Oehlers[1,2,8]

Pathogenic mycobacteria inhibit inflammasome activation to establish infection. Although it is known that potassium efflux is a trigger for inflammasome activation, the interaction between mycobacterial infection, potassium efflux, and inflammasome activation has not been investigated. Here, we use *Mycobacterium marinum* infection of zebrafish embryos and *Mycobacterium tuberculosis* infection of THP-1 cells to demonstrate that pathogenic mycobacteria up-regulate the host WNK signalling pathway kinases SPAK and OXSR1 which control intracellular potassium balance. We show that genetic depletion or inhibition of OXSR1 decreases bacterial burden and intracellular potassium levels. The protective effects of OXSR1 depletion are at least partially mediated by NLRP3 inflammasome activation, caspase-mediated release of IL-1*β*, and downstream activation of protective TNF-*α*. The elucidation of this druggable pathway to potentiate inflammasome activation provides a new avenue for the development of host-directed therapies against intracellular infections.

## Introduction

Inflammasomes are large cytosolic multi-protein complexes that are critical for the immune response to infection. They facilitate the production of bioactive IL-1*β* (Wawrocki & Druszczynska, 2017), which directs pathogen killing through up-regulation of TNF-*α* (Jayaraman et al, 2013) and orchestrates systemic immune control through paracrine signalling. Inflammasomes typically consist of a sensor protein, which detects specific stimuli within the cytosol, and an adaptor protein which facilitates the oligomerization of the sensor with pro–caspase-1 (Taxman et al, 2010; Lu & Wu, 2015). Inflammasome assembly triggers activation of caspase-1, which then cleaves pro-IL-1*β* and pro-IL-18 into their active forms.

Caspase-1 also cleaves Gasdermin D, the N-terminal fragment of which forms pores in the cell membrane, allowing secretion of active IL-1*β* and IL-18, and triggering cell death via pyroptosis (Lu & Wu, 2015). These events contribute to host defence by both rapidly inducing the inflammatory response and limiting replication of intracellular pathogens.

To escape this immune control, many successful intracellular pathogens have evolved methods to limit inflammasome activation (Taxman et al, 2010). Influenza A, *Pseudomonas aeruginosa*, Baculovirus, Vaccinia virus, *Streptococcus pneumoniae*, Myxoma virus, and *Yersinia pseudotuberculosis* have all been shown to limit IL-1*β* production by inhibiting caspase-1 activation (Taxman et al, 2010). In the case of pathogenic mycobacteria, their interactions with inflammasome activation are more complex. One study has shown that *Mycobacterium tuberculosis* actively inhibits inflammasome activation via a zinc metalloprotease (Master et al, 2008), and that clinical isolates associated with severe disease evade NLRP3 activation (Sousa et al, 2020). Others have shown that *M. tuberculosis* induces both NLRP3 inflammasome and caspase-1 activation (Koo et al, 2008; Kurenuma et al, 2009; Dorhoi et al, 2012). Further studies suggest that *M. tuberculosis* actively inhibits activation of the AIM2 inflammasome and dampens activation of the NLRP3 inflammasome by up-regulation of NOS and IFN-*β* (Carlsson et al, 2010; Dorhoi et al, 2012; Briken et al, 2013; Mishra et al, 2013; Wawrocki & Druszczynska, 2017).

The NLRP3 inflammasome, one of the best studied inflammasomes, can be triggered by numerous stimuli, including ATP, heme, pathogen-associated RNA, and a variety of bacterial components (Gupta et al, 2014; Li et al, 2014; Eigenbrod & Dalpke, 2015; Greaney et al, 2015; Erdei et al, 2018). Because these triggers are so diverse, it has been suspected that these activation stimuli are not detected by NLRP3 directly, but rather NLRP3 activation is the result of converging cellular signals. There is evidence that mitochondrial dysfunction, reactive oxygen species, and lysosomal damage contribute to NLRP3 activation (as reviewed elsewhere [He et al,

[1]Tuberculosis Research Program Centenary Institute, The University of Sydney, Camperdown, Australia   [2]The University of Sydney, Discipline of Infectious Diseases and Immunology and Sydney Institute for Infectious Diseases, Camperdown, Australia   [3]Centre for Inflammation and University of Technology Sydney, Faculty of Science, School of Life Sciences, Sydney, Australia   [4]Centenary Imaging and Sydney Cytometry at the Centenary Institute, The University of Sydney, Camperdown, Australia   [5]Epigenetics and RNA Biology Program Centenary Institute, The University of Sydney, Camperdown, Australia   [6]The University of Sydney, Faculty of Medicine and Health, Camperdown, Australia   [7]Department of Clinical Immunology, Royal Prince Alfred Hospital, Camperdown, Australia   [8]A*STAR Infectious Diseases Labs (A*STAR ID Labs), Agency for Science, Technology and Research (A*STAR), Singapore, Singapore

Correspondence: e.hortle@centenary.org.au; stefan_oehlers@idlabs.a-star.edu.sg

2016; Kelley et al, 2019]). A common event that occurs downstream of almost every NLRP3 stimulus is potassium (K$^+$) efflux. Studies have shown that K$^+$ ionophores stimulate NLRP3 (Perregaux & Gabel, 1994), that high extracellular K$^+$ can inhibit NLRP3 activation (Franchi et al, 2007; Pétrilli et al, 2007), and that K$^+$ efflux alone is sufficient to activate the NLRP3 inflammasome (Muñoz-Planillo et al, 2013). Evidence suggests that K$^+$ efflux occurs upstream of NLRP3 activation and induces a conformational change in NLRP3 that favours oligomerization (Meng et al, 2009; Muñoz-Planillo et al, 2013; Tapia-Abellan et al, 2021). This raises the possibility that K$^+$ efflux pathways could be targeted to therapeutically activate, or potentiate the activation of, NLRP3 to control intracellular pathogens.

One of the master regulators of cellular K$^+$ flux is the With-No-Lysine (WNK) kinase signalling pathway. In response to cellular stress or osmotic changes, WNK kinase activates the SPAK and OXSR1 kinases. SPAK and OXSR1 inhibit the KCC channels, which pump K$^+$ out of the cell, and activate the NKCC channels, which pump K$^+$ into the cell (Gagnon & Delpire, 2012). It has previously been shown that blocking the interaction of SPAK/OXSR1 and KCC1 leads to net K$^+$ efflux from the cell (Brown et al, 2015). Indeed, a contemporary article has demonstrated WNK1 deficiency potentiates NLRP3 inflammasome activation in mice (Mayes-Hopfinger et al, 2021). We have further shown that constitutively active KCC1 alters the inflammatory response to malaria infection in mice, and that this effect is associated with dramatically increased survival (Hortle et al, 2019b). Here we sought to investigate whether the SPAK/OXSR1 pathway is involved in the host response to mycobacterial infection, and if this pathway could be manipulated as a host-directed therapy against infection.

# Results

## Infection-induced activation of OXSR1 aids the growth of pathogenic mycobacteria

To determine if SPAK and OXSR1 are involved in immunity, we infected zebrafish embryos with *Mycobacterium marinum* and analysed gene expression at 3 days post infection (dpi). Both *stk39* and *oxsr1a* (the zebrafish orthologs of *SPAK* and *OXSR1*, respectively) were significantly up-regulated at 3 dpi compared with uninfected embryos (Fig 1A). This result is consistent with previous data showing *oxsr1a* is up-regulated in *M. marinum*–infected macrophages (Rougeot et al, 2019).

To determine if the up-regulation of *stk39* and *oxsr1a* results in increased bacterial growth, we depleted each kinase individually by CRISPR-Cas9 knockdown and infected the embryos with fluorescent *M. marinum* (Fig S1). *M. marinum* burden was significantly reduced in *oxsr1a*, but not *stk39*, knockdown embryos (Fig 1B and C). To confirm these results, we created a stable *oxsr1a* knockout allele *oxsr1a*$^{syd5}$ (Fig S2). Homozygous, but not heterozygous, *oxsr1a*$^{syd5}$ embryos showed reduced bacterial burden (Fig 1D).

Because SPAK/OXSR1-modulated K$^+$ channels also shuttle Cl$^-$, Na$^{2+}$, and Ca$^{2+}$, this pathway has been studied for its role in hypertension. The small molecule, Compound B, reduces hypertension in animal models by inhibiting WNK phosphorylation of SPAK/OXSR1 (Ishigami-Yuasa et al, 2017), preventing SPAK/OXSR1 activation (Mori et al, 2013). We first determined that 1.8 µM was the maximum dose of Compound B that could be tolerated by zebrafish larvae for 5 d for infection (Table 1). Immersion of *M. marinum*–infected zebrafish embryos in 1.8 µM Compound B immediately after infection replicated the effect of *oxsr1a* knockdown by decreasing bacterial burden (Fig 1E). This concentration of Compound B did not affect the growth of *M. marinum* in axenic culture (Fig 1F).

In contrast, we did not observe an up-regulation of either kinase when embryos were infected with avirulent ΔESX1–*M. marinum*, which, in the absence of the ESX1 secretion complex, cannot escape the macrophage phagocytic vacuole and fails to activate the inflammasome (Smith et al, 2008) (Fig 2A). The burdens of infection with ΔESX1–*M. marinum* were unchanged by either *oxsr1a* knockdown (Fig 2B), *oxsr1a* knockout (Fig 2C) or Compound B treatment at 1.8 µM (Fig 2D).

## The immunomodulatory role of OXSR1 is conserved across species

To determine whether the immunomodulatory role of OXSR1 is conserved across species, we first differentiated WT THP-1 cells with PMA and infected with *M. marinum*. At 1 dpi OXSR1 protein expression was significantly up-regulated in infected cells compared with uninfected cells (Fig 3A), mirroring the increased *oxsr1a* expression observed in infected zebrafish embryos. Infected THP-1 cells also showed a significant increase in intracellular K+ concentration (Fig 3B and C). To determine whether this up-regulation would affect bacterial burden, we generated an *OXSR1* knockdown human THP-1 cell line (Figs 3D and S3). Although *OXSR1* knockdown cells showed minimal difference in basal K+ concentration compared with control cells, they showed significantly lower K+ concentration when placed in high K+ media (Fig 3E). This suggests that *OXSR1* knockdown cells have increased K+ efflux during osmotic challenge, and is consistent with the known role of *OXSR1* in preventing K+ efflux through KCC channels.

We next infected our *OXSR1* knockdown THP-1 cells with *M. marinum* and *M. tuberculosis* H37Rv and quantified bacterial growth by CFU recovery. At 1 dpi, knockdown THP-1 cells had reduced intracellular *M. marinum* load compared with control THP-1 cells (Fig 3F). At 3 dpi, knockdown THP-1 cells had reduced intracellular *M. tuberculosis* load compared with WT THP-1 cells (Fig 3G). Treatment of infected THP-1 cells with Compound B phenocopied the effect of *OXSR1* knockdown, with reduced *M. marinum* burden at 1 dpi (Fig 3H) and reduced *M. tuberculosis* H37Rv burden at 3 dpi compared with DMSO treatment (Fig 3I).

When we measured K+ concentration in response to WT *M. marinum* infection (Fig S3A), we observed that OXSR1 knockdown THP-1 cells had a much smaller increase in intracellular K+ than control cells (Fig S3B). We also observed that a large proportion of cells were IPG negative, indicating they had very low K+ content. The increase in IPG negative cells was much higher in OXSR1 knockdown cells than in control cells (Fig S3C). Together, these results suggest that OXSR1 knockdown increases K+ efflux during infection. These changes were not observed in ΔESX1 *M. marinum* infection.

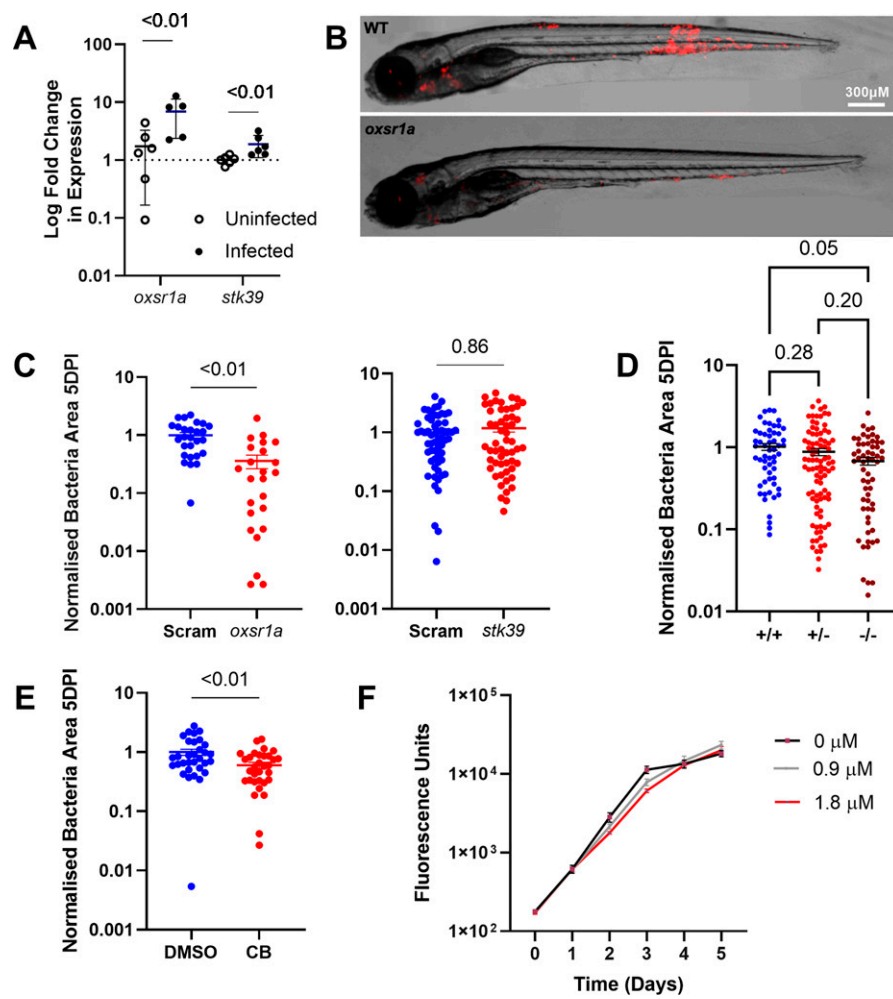

**Figure 1. Infection-induced OXSR1 aids the growth of _M. marinum_ in zebrafish.**
**(A)** Relative expression of _oxsr1a_ and _stk39_ in zebrafish embryos at 3 dpi with WT _M. marinum_, compared with age-matched uninfected controls. Biological replicates (n = 6) each represent pooled RNA from 7 to 10 embryos. **(B)** Representative images of _M. marinum_–tdTomato (red) bacterial burden in WT and mosaic F0 _oxsr1a_ crispant embryos. **(C)** Quantification of _M. marinum_ bacterial burden in WT and mosaic F0 _oxsr1a_ crispant embryos and F0 _stk39_ crispant embryos. Each graph shows combined results of two independent experiments. **(D)** Quantification of _M. marinum_ bacterial burden in WT, heterozygous and homozygous _oxsr1a_ knockout embryos. Graph shows combined results of two independent experiments. **(E)** Quantification of WT _M. marinum_ bacterial burden in DMSO vehicle control and Compound B-treated embryos at 1.8 µM. **(F)** Quantification of axenic _M. marinum_–tdTomato growth by fluorescence in 7H9 broth culture supplemented with Compound B. Red line (1.8 µM) indicates the concentration used to treat infected embryos.

Together, these results indicate that OXSR1 controls cellular potassium flux during mycobacterial infection and that the immunomodulatory role of OXSR1 is conserved between zebrafish and humans.

### Infection-induced OXSR1 suppresses inflammasome activity to aid mycobacterial infection

To determine if the reduced bacterial burden in _oxsr1a_ knockdown was due to increased inflammasome activation, we used CRISPR to knockdown _si:zfos-364h11.1_, a zebrafish protein with orthology to mouse and rat NLRP3, hereafter referred to as _nlrp3_, and the _il1b_ gene which encodes IL-1β (Fig S1). Knockdown of _nlrp3_ alone did not affect the _M. marinum_ burden but ameliorated the protective effect of _oxsr1a_ knockdown against _M. marinum_ infection (Fig 4A and B). The same effect was observed in zebrafish embryos subjected to _il1b_ knockdown in combination with _oxsr1a_ knockdown during _M. marinum_ infection (Fig 4C).

In THP-1 cells infected with _M. tuberculosis_ H37Rv, the _OXSR1_ knockdown-mediated reduction in _M. tuberculosis_ CFU observed at 3 dpi was ameliorated by treatment with the NLRP3 inhibitor MCC950 (Fig 4D), and by NLRP3 knockout (Fig 4E) Supernatant

**Table 1. Compound B toxicity.**

| CB (µM) | % Survival | | |
| --- | --- | --- | --- |
| | 24 h | 48 h | 120 h |
| 0.9 | 100 | 100 | 100 |
| 1.8 | 100 | 100 | 100 |
| 3.6 | 100 | 100 | 47 |
| 7.5 | 100 | 47 | 0 |
| 15 | 5.8 | 0 | 0 |
| 30 | 0 | 0 | 0 |

Survival data of WT zebrafish embryos treated at 1 day post fertilization with varying concentrations of Compound B (CB). Drug was administered once at the beginning of the experiment.

IL-1β was significantly higher in media from _OXSR1_ knockdown cells compared with the WT THP-1 cells after _M. tuberculosis_ H37Rv infection and this increase in IL-1β was ablated by MCC950 treatment (Fig 4F). Together, these data indicate that infection-induced increased expression of _oxsr1a_ increases the mycobacterial burden through suppression of inflammasome activation.

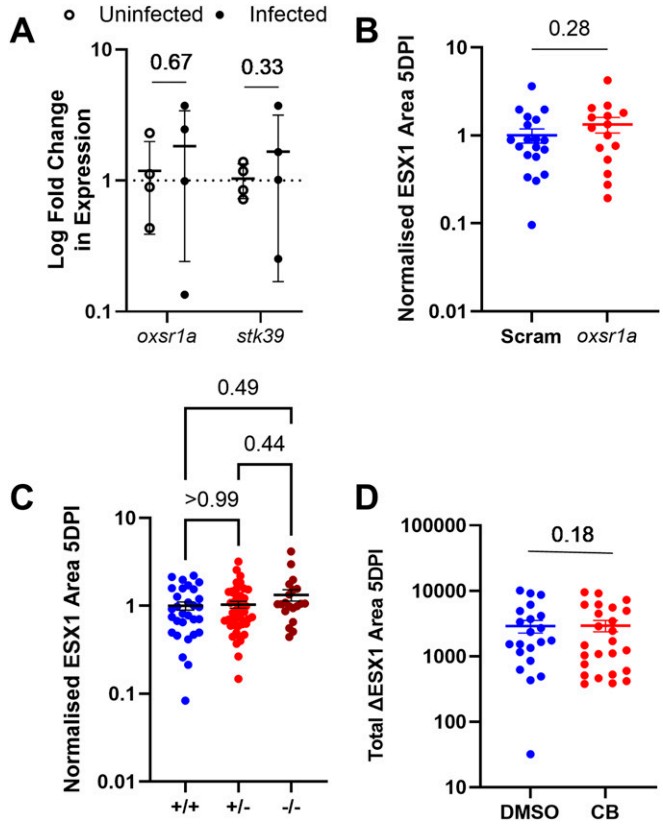

**Figure 2. ΔESX1-*M. marinum* does not induce OXSR1 gene expression.**
**(A)** Relative expression of *oxsr1a* and *stk39* in zebrafish embryos at 3 days post infection with ΔESX1–*M. marinum* compared with age-matched uninfected controls. Biological replicates (n = 4) represent pooled RNA from 7 to 10 embryos. **(B)** Quantification of ΔESX1–*M. marinum* bacterial burden in WT and mosaic F0 *oxsr1a* CRISPR embryos. **(C)** Quantification of ΔESX1–*M. marinum* bacterial burden in WT, heterozygous and homozygous *oxsr1a* knockout embryos. **(D)** Quantification of ΔESX1–*M. marinum* bacterial burden in DMSO vehicle control and Compound B–treated embryos at 1.8 μM.

### Infection-induced OXSR1 suppresses host protective TNF-α and cell death early in infection

Inflammasome-mediated IL-1β increases the macrophage killing of mycobacteria through up-regulation of TNF-α (Jayaraman et al, 2013). We therefore repeated our infection experiments in *TgBAC(tnfa:GFP)^pd1028* embryos to determine if increased TNF-α production was mediating the resistance to mycobacterial infection in *oxsr1a* knockdown zebrafish. The ratio of *tnfa* promoter activity driven GFP per mycobacteria was increased specifically at sites of infection in *oxsr1a* knockdown embryos (Fig 5A) and also in Compound B-treated embryos (Fig 5B). This effect was dependent on *il1b* expression as knockdown of *il1b* suppressed *TgBAC(tnfa:GFP)^pd1028*-driven GFP expression around sites of infection (Fig 5C).

To determine if *tnfa* expression acts downstream of *oxsr1a* depletion, we knocked down *tnfa* expression with CRISPR-Cas9 in the *TgBAC(tnfa:GFP)^pd1028* background to monitor knockdown efficacy (Fig 5D). Knockdown efficacy was also shown by qPCR (Fig S1D). Knockdown of *tnfa* reduced the amount of infection-induced *tnfa*

promoter-driven GFP produced around sites of infection and ameliorated the protective effect of *oxsr1a* knockdown against *M. marinum* infection (Fig 5E). Together, these data suggest increased TNF-α downstream of inflammasome-processed IL-1β is the mechanism driving the lower bacterial burden in *oxsr1a* knockdown embryos.

## Discussion

Here, we used the zebrafish–*M. marinum* and in vitro human–*M. tuberculosis* experimental systems to show that the WNK-OSXR1 signalling pathway potentially has a role in infection-induced activation of the inflammasome. We present evidence that pathogenic mycobacteria increase macrophage K⁺ concentration and induce expression of OXSR1. Infection-induced OXSR1 may suppress protective NLRP3 inflammasome responses and downstream IL-1β/TNF-α production. Several studies have suggested that mycobacteria modulate inflammasome activation either by active inhibition or by up-regulation of NOS, IFN-β, and other negative inflammasome regulators (Briken et al, 2013). Our data expand this literature by showing depletion or inhibition of infection-induced OXSR1 increases host immunity to mycobacterial infection.

In our infection model we found that WT, but not avirulent ΔESX1, *M. marinum* induced expression of both OXSR1 and SPAK. The ESX1 secretion system is essential for the virulence of *M. marinum* and is required for escape of the mycobacteria into the cytoplasm (Conrad et al, 2017; Lienard et al, 2020). This suggests that SPAK/OXSR1 up-regulation is driven by the bacteria and fits with the well-established paradigm that pathogenic mycobacteria co-opt host pathways to establish persistent infection (Oehlers et al, 2015, 2017; Johansen et al, 2018; Hortle et al, 2019a; Hortle & Oehlers, 2020).

We found *stk39* knockdown had no effect on host control of mycobacterial infection in zebrafish embryos. Previous studies have shown that mouse SPAK can play a role in activating macrophage inflammation in both lung injury and inflammatory bowel disease models (Yan et al, 2011; Zhang et al, 2013; Hung et al, 2020). These data raise the possibility that SPAK and OXSR1 may have species or organ specific roles in innate immunity and may respond differently to sterile and infectious triggers of inflammation. It is also likely that OXSR1 interacts with intracellular infections independently of NLRP3 inflammasome (Mayes-Hopfinger et al, 2021), potentially through the direct control of intracellular and vacuolar ion concentrations.

The effect of Compound B on mycobacterial growth showed that small-molecule inhibition of OXSR1 activity can reproduce the impact of OXSR1 knockdown on mycobacterial survival. This observation provides proof of concept that OXSR1 may be a suitable target for host-directed therapies against mycobacterial and other intracellular infections. The reduction in bacterial burden was not as large in Compound B-treated fish as the reduction observed in OXSR1 knockdown embryos. This result may be because the maximum tolerated dose of Compound B was 1.8 μM, which is low compared with some reported $EC_{50}$ values (Mori et al, 2013). Therefore, Compound B may not have reduced OXSR1 activity to the same extent as in the OXSR1 knockdown embryos.

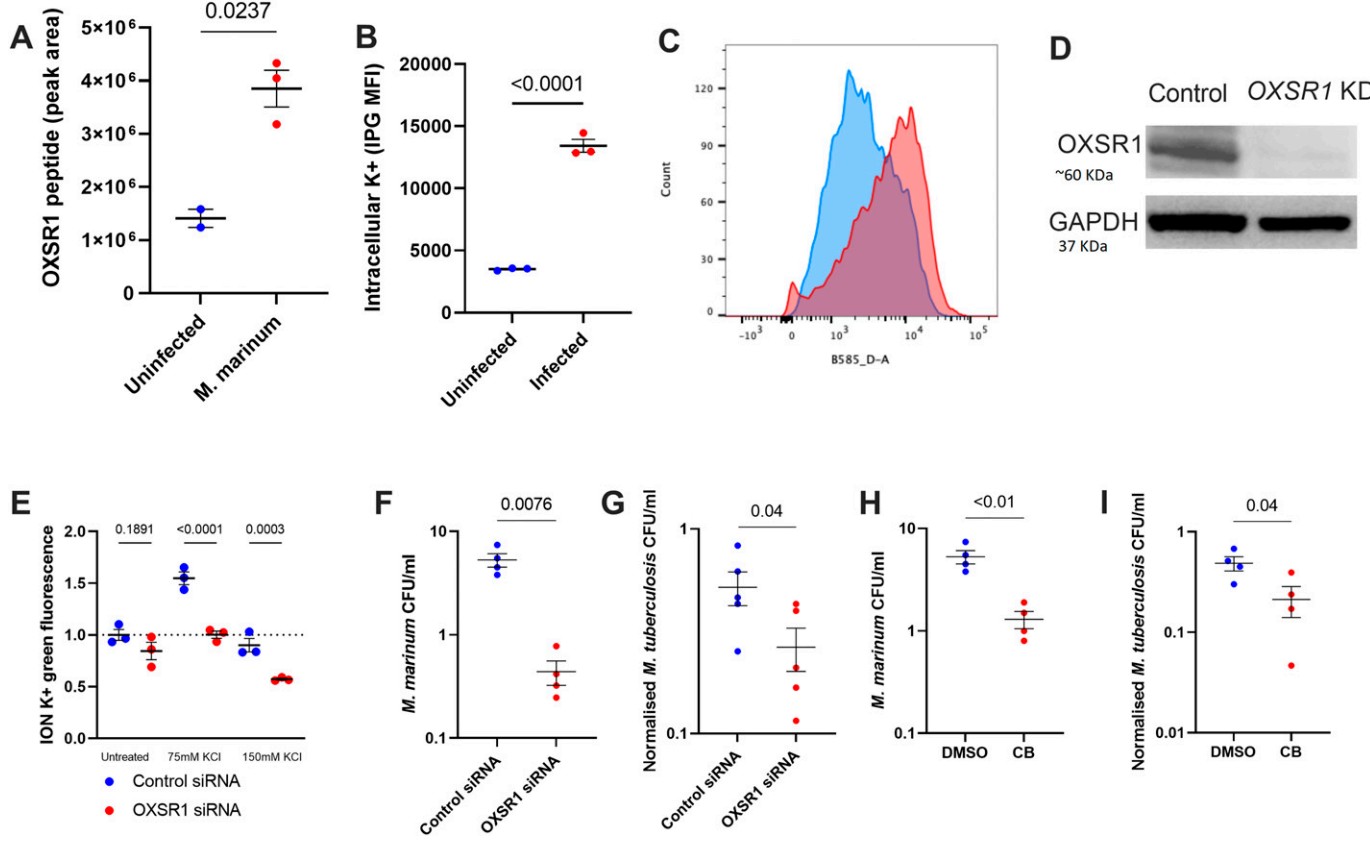

**Figure 3. OXSR1 aids the growth of *M. tuberculosis* in human THP-1 cells.**
**(A)** Quantification of OXSR1 protein in *M. marinum*–infected WT differentiated THP-1 cells, measured by mass spectrometry. **(B)** ION K+ green mean fluorescence intensity (measured by flow cytometry) of *M. marinum*–infected THP-1 cells. **(B, C)** Representative flow cytometry plots of (B). **(D)** Western blot of *OXSR1* knockdown and vector control THP-1 cell lines, showing loss of OXSR1. Full un-edited blots are included in Source Data. **(E)** Fold change of ION K+ green mean fluorescence intensity (measured by flow cytometry) of undifferentiated control and OXSR1 knockdown THP-1 cells in media with increasing K+ content. **(F)** Quantification of intracellular *M. marinum* burden in WT and *OXSR1* knockdown differentiated THP-1 cells at 1 dpi. **(G)** Quantification of intracellular *M. tuberculosis* burden in WT and *OXSR1* knockdown differentiated THP-1 cells at 3 dpi. **(H)** Quantification of intracellular *M. marinum* burden in WT differentiated THP-1 cells treated with Compound B or vehicle control at 1 dpi. **(I)** Quantification of intracellular *M. tuberculosis* burden in WT differentiated THP-1 cells treated with Compound B or vehicle control at 3 dpi. Each dot represents the CFU from an infected well in a single representative experiment and the experiment was repeated three times. *M. tuberculosis* burden data are presented as CFU adjusted to 0 day post infection intracellular bacterial burden.
Source data are available for this figure.

The results from the THP-1-derived macrophages confirm that the role of OXSR1 in the host response to infection is conserved across species. We showed that infection with WT *M. marinum* decreased the K$^+$ content of OXSR1 knockdown cells but did not significantly affect the K$^+$ content of WT cells. Paired with our data showing that WT *M. marinum* increases expression of *oxsr1a*, this suggests that virulent mycobacteria manipulate the SPAK/OXSR1 pathway to maintain high intracellular K$^+$.

Several studies have shown that K$^+$ concentration can affect mycobacterial growth and dormancy, and that successful colonisation of macrophages relies on the ability of the bacteria to maintain K$^+$ homeostasis (Steel et al, 1999; Salina et al, 2014; MacGilvary et al, 2019). Although here we have examined the effects of K$^+$ efflux on inflammasome activation, it is possible that the high K$^+$ maintained in WT cells also aids mycobacterial growth by helping the bacteria maintain the correct ion homeostasis. The mycobacterial infections in THP-1–derived macrophages revealed that both OXSR1 knockdown and treatment with Compound B resulted in

reduced growth of both *M. marinum* and *M. tuberculosis* H37Rv in the human macrophage cell line. With *M. marinum* the maximum reduction was observed at 1 dpi, whereas with *M. tuberculosis* this was not seen until 3 dpi. This is likely to be due to the differing replication times of both pathogens, which are 7 and 24 h, respectively.

Here we have shown that OXSR1 knockdown can only reduce bacterial burden in zebrafish embryos if NLRP3, IL-1$\beta$, and TNF$\alpha$ are functional. In human cells we have shown that infected OXSR1 knockdown cells release significantly more IL-1$\beta$ into the supernatant and that this is ablated by the NLRP3 inhibitor MCC950. Together this suggests that OXSR1 knockdown reduces bacterial burden via a first step of NLRP3 activation. Previous work in the zebrafish–*M. marinum* model has shown both host detrimental and host beneficial effects of inflammasome activation. Whereas morpholino knockdown of *il1b* has been reported to increase bacterial burden, suggesting that *il1b* plays a host protective role, morpholino knockdown of *caspa* reduced bacterial burden,

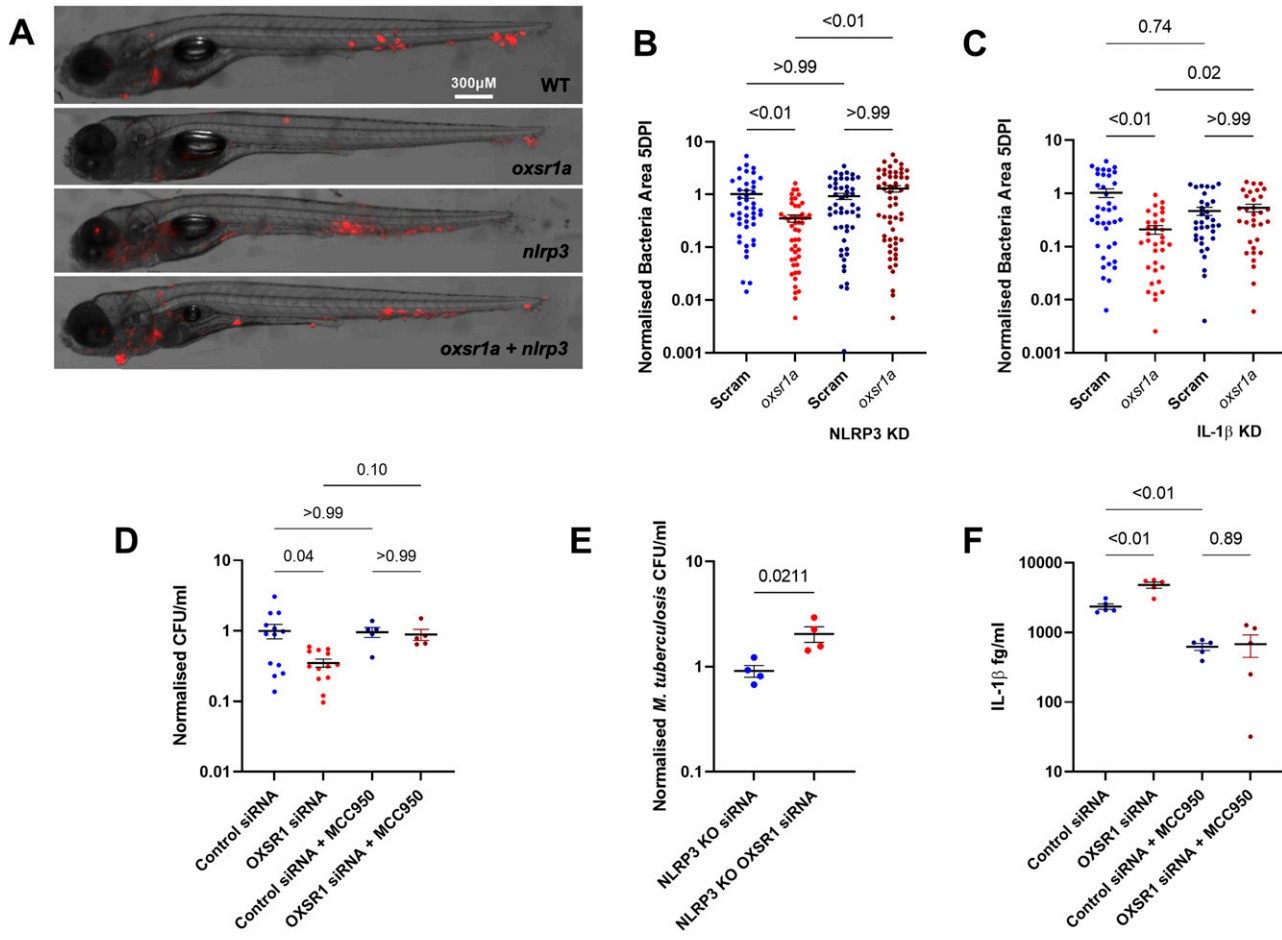

**Figure 4. Infection-induced OXSR1 suppresses inflammasome activity to aid mycobacterial infection.**
**(A)** Representative images of *M. marinum*–tdTomato (red) bacterial burden in 5 dpi WT, mosaic F0 *oxsr1a*, *nlrp3*, and dual *oxsr1a nlrp3* crispant embryos. **(B)** Quantification of WT *M. marinum* bacterial burden in WT, mosaic F0 *oxsr1a*, *nlrp3*, and dual *oxsr1a nlrp3* crispant embryos. Combined results of three biological replicates. **(C)** Quantification of WT *M. marinum* bacterial burden in WT, mosaic F0 *oxsr1a*, *Il1b*, and dual *oxsr1a Il1b* crispant embryos. Combined results of two biological replicates. **(D)** Quantification of intracellular *M. tuberculosis* bacterial burden in WT and O*XSR1* knockdown differentiated THP-1 cells at 3 dpi. **(E)** Quantification of intracellular *M. tuberculosis* burden in WT and *OXSR1* knockdown differentiated NLRP3 knockout THP-1 cells at 3 dpi. **(F)** IL-1β content in the supernatant of *M. tuberculosis*–infected WT and O*XSR1* knockdown differentiated THP-1 cells at 3 dpi. For cell experiments (D, E, F): each dot represents the CFU from an infected well in a single representative experiment and the experiment was repeated three times.

suggesting caspase-associated cell death of infected macrophages benefits the bacteria (Varela et al, 2019 *Preprint*). In our experiments, we did not find any effect of *il1b* or *nlrp3* knockdown on bacterial burden compared with control embryos. This may have been because we were using mosaic F0 CRISPR knockout, which is not a complete removal or because of *M. marinum* strain differences between studies.

We found either Compound B treatment or *oxsr1a* knockdown results in localised increased TNFα production at sites of infection. The fact that we only observed increased TNFα localised to sites of infection, and not throughout the whole embryo, suggests that *oxsr1a* knockdown primes cells for NLRP3 activation but does not cause excess systemic inflammation. Full activation of NLRP3 requires both a priming signal, to induce transcription of NLRP3 components, and pro-IL-1β and an activation signal, to induce oligomerization of NLRP3 (He et al, 2016; Kelley et al, 2019). K⁺ efflux should provide only the second signal (Muñoz-Planillo et al, 2013;

Kelley et al, 2019); therefore, in cells which have not been primed by infection with bacteria we would not expect to see significant NLRP3 activation. This suggests that OXSR1 inhibition may be an effective host-directed therapy strategy that induces beneficial inflammation at sites of infection without inducing detrimental systemic inflammation. Any therapy that seeks to increase inflammation must be treated with caution as chronic production of IL-1β and TNFα are likely to be detrimental during chronic infection.

Our findings that OXSR1 can be targeted to decrease bacterial burden define a new avenue for the development of host-directed therapy. Although numerous studies have investigated the potential of inhibiting NLRP3 to minimize pathology (Yang et al, 2019), the possibility of activating inflammasomes to increase pathogen clearance has been largely unexplored. Here we have shown that enhancing inflammasome activation via K⁺ efflux can provide the dual benefits of maximising the anti-pathogen effects of inflammation without causing excess tissue damage. Given mycobacteria

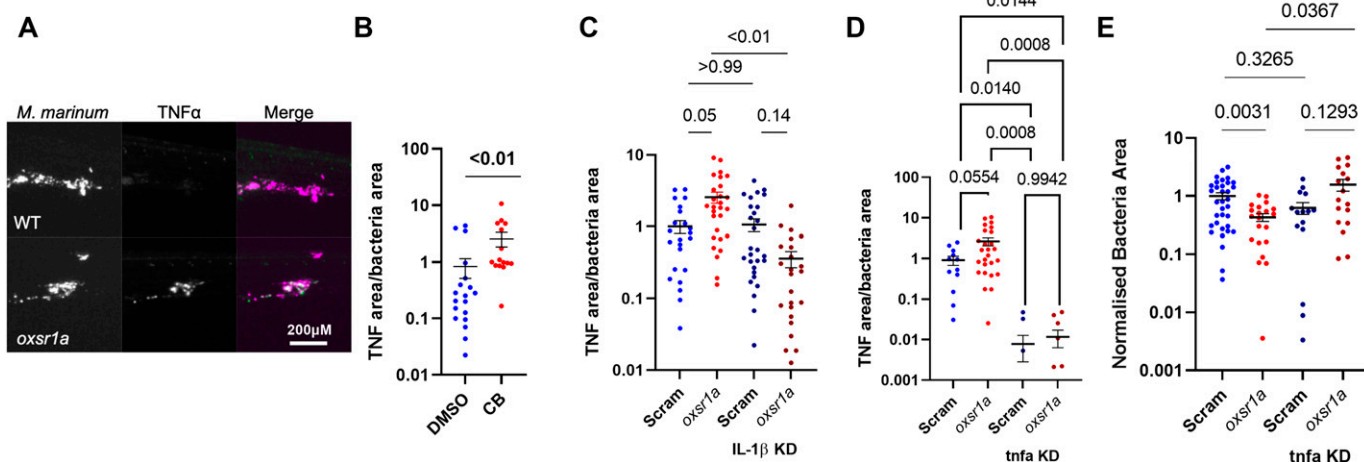

**Figure 5. Infection-induced OXSR1 suppresses inflammasome activity to aid mycobacterial infection.**
**(A)** Representative images of *Tg(tnfa:GFP)*[pd1028] fluorescence around *M. marinum* granulomas. Scale bar represents 200 μM. **(B)** Quantification of TNF-α fluorescence per bacterial area in WT embryos treated with Compound B. **(C)** Quantification of TNF-α fluorescence per bacterial area in WT, mosaic F0 *oxsr1a*, *Il1b*, and dual *oxsr1a Il1b* crispant embryos. Combined results of two independent experiments. **(D)** Quantification of TNF-α fluorescence per bacterial area in WT, mosaic F0 *oxsr1a*, *tnfa*, and dual *oxsr1a tnfa* crispant embryos. Combined results of two independent experiments. **(E)** Quantification of bacterial burden in WT, mosaic F0 *oxsr1a*, *tnfa*, and dual *oxsr1a tnfa* crispant embryos. Combined results of two independent experiments. Fish were infected with WT *M. marinum* by caudal vein injection at 30 h post fertilization and analysed 5 dpi.

are not the only pathogens which inhibit inflammasome activation, OXSR1 inhibition may be an effective host-directed therapy with broad applicability.

# Materials and Methods

## Zebrafish husbandry

Adult zebrafish were housed at the Centenary Institute (Sydney Local Health District AWC Approval 2017-036). Zebrafish embryos were obtained by natural spawning and embryos were raised at 28°C in E3 media.

## Zebrafish lines

Wild type zebrafish are the TAB background. Transgenic line was *Tg(tnfa:GFP)*[pd1028] (Marjoram et al, 2015).

## Infection of zebrafish embryos

Embryos were infected by microinjection with ~400 fluorescent *M. marinum* M strain and ΔESX1 *M. marinum* as previously described (Matty et al, 2016). Embryos were recovered into E3 supplemented with 0.036 g/l PTU, housed at 28°C, and imaged on day 5 of infection unless otherwise stated.

## Quantitative (q)RT-PCR

RNA was extracted from 5 to 10 embryos using TRIzol (Invitrogen) according to the manufacturer's instructions. Equal amounts of RNA (either 1 or 2 μg depending on RNA yield) were used for the cDNA synthesis reaction. qRT-PCR reactions were carried out on a

**Table 2. Primers used for gene expression studies.**

| | Sequence 5′-3′ |
|---|---|
| Oxsr1a qFw | gctgctttacggtcaccaag |
| Oxsr1a qRv | attttagccgagtcctgccc |
| Stk39 qFw | gatcgcagattttggcgtga |
| Stk39 qRv | gatatcgatggtacggcgca |
| EF1a qFw | tgccttcgtcccaatttcag |
| EF1a qRv | taccctccttgcgctcaatc |
| IL-1β qFW | atcaaaccccaatccacagagt |
| IL-1β qRv | ggcactgaagacaccacgtt |

Biorad CFX machine using Thermo Fisher Scientific PowerUP SYBR green and primers described in Table 2. The relative quantity of transcripts was calculated by the delta–delta CT method.

## Imaging

Live zebrafish embryos were anaesthetized in M-222 (Tricaine) and mounted in 3% methylcellulose for static imaging on a Leica M205FA fluorescence stereomicroscope. Fluorescent pixel count analyses were carried out with Image J Software Version 1.51j and intensity measurements were performed as previously described (Matty et al, 2016).

## CRISPR-Cas9 knockdown and mutant generation

Primers used for gRNA transcription are detailed in Table 3 and were designed by Wu et al (2018). Templates for gRNA transcription were produced by annealing and amplifying gene specific oligos to the scaffold oligo using the NEB Q5 polymerase. Pooled

**Table 3.** Primers used to generate gRNAs for CRISPR-Cas9 knockdown.

| | Sequence 5'-3' |
|---|---|
| il1b_Target_1 | TAATACGACTCACTATAGGGTTCAGATCCGCTTGCAAGTTTTAGAGCTAGAAATAGC |
| il1b_Target_2 | TAATACGACTCACTATAGGCATGGCGAACGTCATCCAGTTTTAGAGCTAGAAATAGC |
| il1b_Target_3 | TAATACGACTCACTATAGGCACTGGGCGACGCATACGGTTTTAGAGCTAGAAATAGC |
| il1b_Target_4 | TAATACGACTCACTATAGGCAGCTGGTCGTATCCGTTGTTTTAGAGCTAGAAATAGC |
| oxsr1a_Target_1 | TAATACGACTCACTATAGGGTTGAGAGCTCGGGTCCTGTTTTAGAGCTAGAAATAGC |
| oxsr1a_Target_2 | TAATACGACTCACTATAGGGCACCTCTCTTAGTATGGGTTTTAGAGCTAGAAATAGC |
| oxsr1a_Target_3 | TAATACGACTCACTATAGGTCCAGTCTCTAAACACGGGTTTTAGAGCTAGAAATAGC |
| oxsr1a_Target_4 | TAATACGACTCACTATAGGAGGCGGTGCCGAATGCGGGTTTTAGAGCTAGAAATAGC |
| scramble_target_1 | TAATACGACTCACTATAGGCAGGCAAAGAATCCCTGCCGTTTTAGAGCTAGAAATAGC |
| scramble_target_2 | TAATACGACTCACTATAGGTACAGTGGACCTCGGTGTCGTTTTAGAGCTAGAAATAGC |
| scramble_target_3 | TAATACGACTCACTATAGGCTTCATACAATAGACGATGGTTTTAGAGCTAGAAATAGC |
| scramble_target_4 | TAATACGACTCACTATAGGTCGTTTTGCAGTAGGATCGGTTTTAGAGCTAGAAATAGC |
| si:zfos-364h11.1_Target_1 | TAATACGACTCACTATAGGTATAGAGACTCTTTGTACGTTTTAGAGCTAGAAATAGC |
| si:zfos-364h11.1_Target_2 | TAATACGACTCACTATAGGGATCTGATTAGTTGCTGCGTTTTAGAGCTAGAAATAGC |
| si:zfos-364h11.1_Target_3 | TAATACGACTCACTATAGGGCTTCGTCACTGAATTCAGTTTTAGAGCTAGAAATAGC |
| si:zfos-364h11.1_Target_4 | TAATACGACTCACTATAGGAGCTCTCTTAGTGAGTTTGTTTTAGAGCTAGAAATAGC |
| STK39_Target_1 | TAATACGACTCACTATAGGGTAGTAGGTGACCACGTTGTTTTAGAGCTAGAAATAGC |
| STK39_Target_2 | TAATACGACTCACTATAGGGACCTGCTCCATTACTTCGTTTTAGAGCTAGAAATAGC |
| STK39_Target_3 | TAATACGACTCACTATAGGAGAACGATCCTCCCTCGCGTTTTAGAGCTAGAAATAGC |
| STK39_Target_4 | TAATACGACTCACTATAGGCAGGTGTCCACTCGACCCGTTTTAGAGCTAGAAATAGC |
| tnfa_Target_1 | TAATACGACTCACTATAGGTTGAGAGTCGGGCGTTTTGTTTTAGAGCTAGAAATAGC |
| tnfa_Target_2 | TAATACGACTCACTATAGGTCTGCTTCACGCTCCATAGTTTTAGAGCTAGAAATAGC |
| tnfa_Target_3 | TAATACGACTCACTATAGGGATTATCATTCCCGATGAGTTTTAGAGCTAGAAATAGC |
| tnfa_Target_4 | TAATACGACTCACTATAGGTCCTGCGTGCAGATTGAGGTTTTAGAGCTAGAAATAGC |
| Scaffold | AAAAGCACCGACTCGGTGCCACTTTTTCAAGTTGATAACGGACTAGCCTTATTTTAACTTGCTATTTCTAGCTCTAAAAC |

transcription of gRNAs was carried out using the NEB HiScribe T7 High Yield RNA Synthesis Kit.

Embryos were injected at the single cell stage with an injection mix containing 1 μl phenol red, 2 μl 500 ng/μl pooled guides, and 2 μl of 10 μM Cas9. All "Scram" embryos are injected with scrambled guide RNA.

To create oxsr1a knockout line, F0 crispants were outcrossed to WT AB, and HRM analysis was conducted on F1 progeny. F1s with a visible HRM shift using primers amplifying the four predicted cut sites (primer 3 spanned two cut sites) were sent for sanger sequencing (Table 4). An F1 zebrafish was discovered carrying an 8-bp deletion predicted to cause a premature stop at amino acid 13 (Fig S2). F2 progeny were genotyped with a custom KASP assay ordered from LGC Biosearch Technologies.

## Drug treatments

Embryos and cells were treated with vehicle control (DMSO or water as appropriate), 10 μM MCC950, or 1.8 μM Compound B (Stock2S 26016 Tocris) immediately after infection. For zebrafish, the drugs

**Table 4.** Genotyping primers.

| | Sequence 5'-3' |
|---|---|
| Oxsr1a Fw 1 | Aagtttggctgttgggactg |
| Oxsr1a Rv 1 | Agatgctgatgtgtggtgga |
| Oxsr1a Fw 2 | ctgttttcagGCTCAGTGCTT |
| Oxsr1a Rv 2 | TCCAGCCTTCACATCCctac |
| Oxsr1a Fw 3-4 | Gtttttccacagttctggtttt |
| Oxsr1a Rv 3-4 | Ccttctggaggcacaaagag |

and E3 were replaced on days 0, 2, and 4 dpi. For cell culture, drugs were replaced at 4 h post infection.

## Axenic culture

A mid-log culture of fluorescent *M. marinum* was diluted 1:100 and aliquoted into 96-well plates for drug treatment. Cultures were maintained at 28°C in a static incubator and bacterial fluorescence was measured in a BMG Fluorostar plate reader.

### THP-1 cell culture

Human THP-1 cells (ATCC TIB-202) were cultured in RPMI media (22400089; Thermo Fisher Scientific) supplemented with 1% (vol/vol) non-essential amino acids (11140050; Thermo Fisher Scientific), 1 mM sodium pyruvate (11360070; Thermo Fisher Scientific), 10% (vol/vol) FCS (Hyclone; GE Healthcare), and 0.1 mg/ml penicillin/streptomycin (15140122; Thermo Fisher Scientific) at 37°C, 5% $CO_2$.

NLRP3 knockout THP-1 cells were purchased from InvivoGen (catalog code thp-konlrpez), and cultured according to the manufacturer's instructions before transduction with virus.

### Viral production

24 h before transfection, $4 \times 10^6$ HEK2937 cells were seeded in a 100 mm culture dish. On the day of transfection, cells were co-transfected with 15 $\mu$g of the pLKO.1_GFP (#30323; Addgene) vector containing OXSR1_Sh1, OXSR_Sh2 or AthmiR, 6.5 $\mu$g of the packaging plasmid pMDL-g/prre (#12251; Addgene), 2.5 $\mu$g of the packaging plasmid pRSV-Rev (#12253; Addgene), and 3.5 $\mu$g of the envelop expressing plasmid pMD2-VSV-G (#12259; Addgene) by the calcium phosphate transfection method. Culture media was changed the following day and cells were cultured for another 24 h. Medium containing lentiviral particles was then collected, debris was cleared by centrifugation at 430$g$ for 5 min, filtered through a 0.45-$\mu$m filter, aliquoted, and stored at –80°C.

### Transduction

Briefly, $5 \times 10^5$ THP-1 cells were resuspended in 500 $\mu$l of fresh culture media containing 10 $\mu$g/ml polybrene. After adding 50 $\mu$l of virus, cells were spinoculated for 90 min at 462$g$, 22°C. After spinning, pelleted cells were resuspended in in the same media and incubated for 4 h at 37C, 5% $CO_2$. After incubation, the cells were pelleted, resuspended in fresh culture media and transferred to a six-well plate. The cells were cultured for 48 h before FACS selection.

### Western blotting

Protein lysates were loaded onto 4–12% BIS-Tris Protein gels (NP0336BOX; Thermo Fisher Scientific) for electrophoresis followed by transfer onto a PVDF membrane (MILIPVH00010; Merck Millipore). Membrane was blocked with 5% (vol/vol) skim milk for 1 h at room temperature, incubated overnight with a 1:1,000 dilution of rabbit anti-OXSR1 (ab97694; Abcam) followed by incubation with a 1:5,000 dilution of a donkey anti-rabbit IgG HRP antibody (AP182P; Merck Millipore) and 1:5,000 dilution of mouse anti-GAPDH (ab8245; Abcam) antibody, followed by incubation with a 1:5,000 dilution of a donkey anti-mouse IgG HRP antibody (AP192P; Merck Millipore). Protein detection was performed using SuperSignal West Pico PLUS (34579; Thermo Fisher Scientific) and imaged on a Bio-Rad ChemiDoc Imaging System.

### Mass spectrometry

LCMS grade reagents obtained from Sigma-Aldrich included: acetonitrile (ACN), TFA, formic acid (FA), sodium deoxycholate (SDC), Tris–HCl pH 8.0, chloroacetamide (CLA), and Tris(2-carboxyethyl)-phosphine (TCEP). Trypsin gold was purchased from Promega and sulfonated divinylbenzene (SDB-RPS) discs were purchased from Affinsep. "MasterMix" consists of: 1% SDC, 10 mM TCEP, 40 mM CLA, and 100 mM tris and was made in bulk, aliquoted and frozen. Appropriate amounts were thawed before use.

Samples were prepared as described previously (O'Rourke et al, 2021) with some modifications: Cell samples suspended in RIPA buffer were first normalized and 50 $\mu$l of "MasterMix" was added. Each sample was then boiled at 95°C for 10 min and allowed to cool on the bench. Using the previous quantitation as a guide, Sequencing grade Trypsin was added to a ratio of between 1:50 and 1:100 before being incubated overnight at 37°C (~16 h).

Digested samples were then desalted and delipidated by STAGE-Tip purification using a custom 3D printed STAGE tip Bracket (O'Rourke et al, 2021) and the following method: First dilute samples with 10X volumes of 100% ACN next add samples to STAGE TIPS and spin for 3 min at 1,000$g$ in a centrifuge. Samples were then washed in sequence with 100 $\mu$l of: 90% ACN 1% TFA and 10% ACN 0.1% TFA followed by elution into a fresh 96-well plate with 100 $\mu$l of elution buffer containing 71 $\mu$L 1M NH4OH3, 800 $\mu$L of 100% acetonitrile, and 129 $\mu$L water. Samples were then allowed to dry in ambient conditions in a fume hood until dry.

Dry peptides were re-constituted in 25 $\mu$l of buffer A (0.1% FA) followed by LCMS analysis. Using an Acquity M-class nanoLC system (Waters), 5 $\mu$l of the sample (1 mg) was loaded at 15 ml/min for 3 min onto a nanoEase Symmetry C18 trapping column (180 × 20 mm) before being washed onto a PicoFrit column (75 mmID × 250 mm; New Objective) packed with Magic C18AQ resin (Michrom Bio-resources). Peptides were eluted from the column and into the source of a Q Exactive Plus mass spectrometer (Thermo Fisher Scientific) using the following program: 5–30% MS buffer B (98% Acetonitrile + 0.2% FA) over 90 min, 30–80% MS buffer B over 3 min, 80% MS buffer B for 2 min, and 80–5% for 3 min. The eluting peptides were ionised at 2,000 V. A data-dependant MS/MS (dd-MS2) experiment was performed, with a survey scan of 350–1,500 D performed at 70,000 resolution for peptides of charge state 2+ or higher with an AGC target of $3 \times 10^6$ and maximum Injection Time of 50 ms. The top 12 peptides were selected fragmented in the HCD cell using an isolation window of 1.4 m/z, an AGC target of $1 \times 10^5$ and maximum injection time of 100 ms. Fragments were scanned in the Orbitrap analyser at 17,500 resolution and the product ion fragment masses measured over a mass range of 120–2,000 D. The mass of the precursor peptide was then excluded for 30 s.

The MS/MS data files were searched using Peaks Studio (version 8.5) against the Human Proteome database combined with the *M. marinum* proteome and a database of common contaminants with the following parameter settings. Fixed modifications: none. Variable modifications: oxidised methionine and deamidated asparagine carbamidomethylation of cystine. Enzyme: trypsin. Number of allowed missed cleavages: 3. Peptide mass tolerance: 30 ppm. MS/MS mass tolerance: 0.1 D. Charge state: 2+, 3+, and 4+. The results of the search were then filtered to include peptides with a $-\log_{10}P$ score that was determined by the false discovery rate (FDR) of <1%, the score being that where decoy database search matches were <1% of the total matches. LFQ calculations were also performed as part of the workflow using an FDR of 1% with the reference samples being the uninfected control group.

### ION K+ green (undifferentiated cells)

For flow cytometry, $2 \times 10^5$ undifferentiated THP-1 cells/well (ATCC TIB-202) were seeded into a 96 well plate and incubated at 37°C for 1.5–2 h with either furosemide, 37.5 mM or 75 mM KCl. ION potassium green-2 AM (ab142806; Abcam) was added to a final concentration of 52.8 mM and cells were incubated for a further 15 min. Cells were spun down for 5 min at 462$g$ and resuspended in PBS + 2% FCS supplemented with either furosemide, 37.5 mM or 75 mM KCl. ION K+ green fluorescence was captured on a BD Fortessa through the PE channel. 5,000 events were captured per sample.

### Proteomics

Samples were prepared as previously described (O'Rourke et al, 2021), with minor modifications (Appendix Supplementary Methods). The MS/MS data files were searched using Peaks Studio (version 8.5) against the Human Proteome database combined with the *M. marinum* proteome and a database of common contaminants with the following parameter settings. Fixed modifications: none. Variable modifications: oxidized methionine and deamidated asparagine carbamidomethylation of cystine. Enzyme: tyrypsin. Number of allowed missed cleavages: 3. Peptide mass tolerance: 30 ppm. MS/MS mass tolerance: 0.1 D. Charge state: 2+, 3+ and 4+. The results of the search were then filtered to include peptides with a $-\log_{10}P$ score that was determined by the FDR of <1%, the score being that where decoy database search matches were <1% of the total matches. LFQ calculations were also performed as part of the workflow using an FDR of 1% with the reference samples being the uninfected control group. The mass spectrometry proteomics data have been deposited to the ProteomeXchange Consortium via the PRIDE (Perez-Riverol et al, 2019) partner repository with the dataset identifier PXD030631.

### ION K+ green staining

$2 \times 10^5$ THP-1 cells/well were seeded into a 96-well plate and differentiated for 24 h with 100 mM PMA. Cells were then infected with frozen single cell preparation *M. marinum*–Katushka at an MOI of 1. After 4 h, extracellular bacteria were removed by washing with PBS + 2% FCS, and cells were incubated at 32°C for 3 d. Cells were lifted from the plate by 15 min incubation at 37°C with Accutase (StemCell Technologies), then washed with PBS + 2% FCS. ION K+ green was added to a final concentration of 52.8 mM and cells were incubated for a further 15 min. Cells were spun down for 5 min at 462$g$ and resuspended in PBS + 2% FCS for flow cytometry. ION K+ Green fluorescence was captured on a BD Fortessa through the FITC channel (so as not to overlap with Katushka). 5,000 events were captured per sample.

### Mycobacterial infection of THP-1 cells

$2 \times 10^5$ THP-1 cells/well were seeded into a 96 well plate and differentiated for 24 h with 100 mM PMA. Cells were then infected with either mid log culture of *M. tuberculosis* H37Rv or frozen single cell preparation of *M. marinum* at an MOI of 1. After 4 h, extracellular bacteria were removed by washing with PBS + 2% FCS, and cells were incubated at either 32°C for *M. marinum* infections or 37°C for *M. tuberculosis* infections.

### Mycobacterial CFU recovery from THP-1 cells

Cells were washed in PBS + 2% FCS and lysed with TDW + 1% Triton X100 for 10 min. Lysate was serially diluted and plated on 7H10 agar supplemented with 50 $\mu$g/ml hygromycin for the recovery of *M. marinum* or a mix of 200,000 units/l polymyxin B, 50 mg/l carbenicillin, 10 mg/l amphotericin B, and 20 mg/l trimethoprim lactate for the recovery of *M. tuberculosis*. Plates were incubated at 32°C for 7 d (*M. marinum*) or 37°C for 14 d (*M. tuberculosis*).

### Measurement of human IL-1$\beta$ in supernatants

IL-1$\beta$ was measured by cytometric bead array, using a human IL-1$\beta$ enhanced-sensitivity flex set (BD Biosciences). Undiluted cell supernatant was stained according to the manufacturer's instructions and run on a BD FACS Canto II. Data were analysed using FCAP array software.

### Statistics

All statistical tests were calculated in GraphPad Prism. $T$ tests were unpaired $t$ tests with Welch's correction. All ANOVA were ordinary one-way ANOVA, comparing the means of specified pairings, using Turkey's multiple comparisons test with a single pooled variance. In cases where data were pooled from multiple experiments, data from each were normalized to its own within-experiment control (usually DMSO) before pooling. Error bars indicate SEM. Outliers were removed using ROUT, with Q = 1%.

## Data Availability

The mass spectrometry proteomics data have been deposited to the ProteomeXchange Consortium via the PRIDE partner repository with the dataset identifier PXD030631 (Perez-Riverol et al, 2019).

## Supplementary Information

## Acknowledgements

We thank Dr Kristina Jahn of Sydney Cytometry for assistance with imaging equipment, Ms Kaiming Luo and Dr Pradeep Cholan for technical assistance, and all members of the Tuberculosis Research Program at the Centenary Institute for helpful comments. This work was supported by the Australian National Health and Medical Research Council (grant numbers APP1099912, APP1053407 to SH Oehlers; APP1153493 to WJ Britton); University of Sydney Fellowship (grant number G197581 to SH Oehlers); NSW Ministry of Health under the NSW Health Early-Mid Career Fellowships Scheme (grant number H18/31086 to SH Oehlers); the Kenyon Family Inflammation Award (2019 to E Hortle); and the Centenary Institute Booster Grant (2020 to E Hortle).

## Author Contributions

E Hortle: conceptualization, data curation, formal analysis, supervision, funding acquisition, investigation, visualization, methodology, project administration, and writing—original draft, review, and editing.
VLT Tran: data curation, formal analysis, and investigation.
K Wright: formal analysis, validation, and investigation.
ARM Fontaine: formal analysis, visualization, and methodology.
N Pinello: investigation and methodology.
MB O'Rourke: resources, data curation, software, formal analysis, investigation, and methodology.
JJL Wong: investigation and methodology.
PM Hansbro: resources.
WJ Britton: resources, formal analysis, supervision, funding acquisition, and writing—review and editing.
SH Oehlers: conceptualization, resources, data curation, formal analysis, supervision, funding acquisition, validation, investigation, visualization, methodology, project administration, and writing—original draft, review, and editing.

## Conflict of Interest Statement

The authors declare that they have no conflict of interest.

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
