## [Reviewer comments · Life Science Alliance]

Life Science Alliance

OXR1 inhibits inflammasome activation by limiting potassium efflux during mycobacterial infection

Elinor Hortle, Vi Tran, Kathryn Wright, Angela Fontaine, Natalia Pinello, Matthew O'Rourke, Justin Wong, Philip Hansbro, Warwick Britton, and Stefan Oehlers

DOI: <https://doi.org/10.26508/lsa.202201476>

Corresponding author(s): Stefan Oehlers, Agency for Science, Technology and Research and Elinor Hortle, Centenary Institute of Cancer Medicine and Cell Biology

Review Timeline:

Submission Date:	2022-04-08
Editorial Decision:	2022-04-14
Revision Received:	2022-04-15
Accepted:	2022-04-19

Transaction Report:

Please note that the manuscript was reviewed at Review Commons and these reports were taken into account in the decision-making process at Life Science Alliance.

April 14, 2022

RE: Life Science Alliance Manuscript #LSA-2022-01476

Dr. Stefan H Oehlers
Agency for Science, Technology and Research
8A Biomedical Grove, Immunos Building
Level 5
Singapore, NSW 138648

Dear Dr. Oehlers,

Thank you for submitting your revised manuscript entitled "OXSR1 inhibits inflammasome activation by limiting potassium efflux during mycobacterial infection". We would be happy to publish your paper in Life Science Alliance pending final revisions necessary to meet our formatting guidelines.

- please add a Running Title, Category, and an Alternate Abstract / Summary Blurb to our system
- please add the Twitter handle of your host institute/organization as well as your own or/and one of the authors in our system
- please make sure that all authors listed in the manuscript are entered in our system and list the author contributions both in the main manuscript text and our system
- Please incorporate the methods from the Supplemental Material into the main Materials & Methods section of the paper. We have no word limit for this section.
- Figure S1 has a panel D, but this is not mentioned in the figure legend
- please add molecular weights next to all blots
- Figure S3 should be relabeled as Source Data, with a mention of which Figures this data is for

A. FINAL FILES:

B. MANUSCRIPT ORGANIZATION AND FORMATTING:

Sincerely,

April 19, 2022

RE: Life Science Alliance Manuscript #LSA-2022-01476R

Dr. Stefan H Oehlers
Agency for Science, Technology and Research
8A Biomedical Grove, Immunos Building
Level 5
Singapore, NSW 138648
Singapore

Dear Dr. Oehlers,

Thank you for submitting your Research Article entitled "OXSR1 inhibits inflammasome activation by limiting potassium efflux during mycobacterial infection". It is a pleasure to let you know that your manuscript is now accepted for publication in Life Science Alliance. Congratulations on this interesting work.

DISTRIBUTION OF MATERIALS:

Again, congratulations on a very nice paper. I hope you found the review process to be constructive and are pleased with how the manuscript was handled editorially. We look forward to future exciting submissions from your lab.

Sincerely,
